# The Enhanced Measurement Method Based on Fiber Bragg Grating Sensor for Structural Health Monitoring

**DOI:** 10.3390/mi16040368

**Published:** 2025-03-24

**Authors:** Shengtao Niu, Ru Li

**Affiliations:** 1Hangzhou Institute of Technology, Xidian University, Hangzhou 311231, China; 2The School of Mechano-Electronic Engineering, Xidian University, Xi’an 710071, China; 3School of Intelligent Manufacturing and Control Engineering, Qilu Institute of Technology, Ji’nan 250200, China; lllr0312@qlit.edu.cn

**Keywords:** measurement method, structural health monitoring, inverse finite element method, three-dimensional deformation, fiber Bragg grating strain sensors

## Abstract

The effective measurement method plays a vital role in the structural health monitoring (SHM) field, which provides accurate and real-time information concerning structural conditions and performance. The innovative measurement approach based on strain sensors, referred to as the inverse finite element method (iFEM), has been considered the most promising and versatile technology for meeting the requirements of the SHM system. However, the existing iFEM for shape sensing of thick plate structures has the drawback that the transverse shear effect makes no contribution to the three-dimensional deformation of thick plate structures. Therefore, this study proposed an enhanced inverse finite element method (iFEM) based on single-surface fiber Bragg grating strain sensors for reconstructing thick plate structures coupled with an analytical formulation. The method characterized the explicit relationship between transverse shear and bending displacement field on the mid-plane, which presents the sixth-order differential equation based on a variational approach. The three-dimensional deformation field can be obtained along the thickness direction, expanding the SHM application of iFEM for composite structures based on strain measurement. By performing shape sensing analysis of the thick plate model, the exactness and applicability of the present method are numerically and experimentally validated for different loading cases.

## 1. Introduction

The increasing interest in real-time measurement methods has heightened the need for structural health monitoring (SHM) systems in the modern engineering field [1,2]. SHM is considered an interdisciplinary procedure, including structural sensing systems, immediate data collection and processing, and real-time information feedback to control systems about the global or local structural state. The main purpose of SHM is to detect unusual structural behaviors to pinpoint failures or unhealthy structural conditions [3], thereby increasing the safety of humans and the environment while also reducing maintenance costs. However, the real-time measurement of structural three-dimensional displacement fields, which obtains the stress and strain within a given structure, is an instrumental technology for the SHM system of engineering structures.

Existing measurement methodologies from the past decade can largely be grouped as either noncontact means based on the principle of optical imaging or contact-forming, based on strain sensors bonded to or embedded in the structural surface [4]. Compared to contact measurement, noncontact methods are less accurate for the application of large equipment due to interference from the external environment, such as the circumstances of measurement and the surrounding environment. On the contrary, the contact method, employing different types of sensors [5,6,7], such as strain gauge sensors, acceleration sensors, and fiber Bragg grating (FBG) sensors, can obtain the physical information on objective structures and reconstruct the full-field deformation field using the differential and integral relationships and the physical information on objective structures. Moreover, benefiting from the significant advantages of FBG, such as being lightweight and small in size, corrosion-resistant, and resistant to electromagnetic interference [8,9], much research has been devoted to the analytical evaluation and experimental testing of FBG sensors. Additionally, a great deal of previous research based on FBG sensors focuses on displacement measurement; Chen and Hu et al. [10] conducted large-range curvature measurements based on FBGs in two-core fiber with a protective coating, which proved to be a good candidate for curvature sensing in the engineering field. Xu et al. [11] presented an online measurement and calibration method for estimating the deformation of the sub-reflector support structure based on FBG sensors, which eliminates the effects of temperature variations on strain measurements using a temperature-compensating device. Rakotondrabe et al. [12] employed embedded FBG sensors to detect and monitor damage to laminated composite structures, thereby developing a more cost-effective aircraft maintenance approach.

Furthermore, the measurement of structural three-dimensional displacement based on FBG sensors, known as shape sensing, has emerged as a powerful platform for SHM systems. However, shape sensing is related to an inverse problem from a mathematical perspective [13], so the solving process results in inevitable problems, such as the uniqueness of given cases and instability arising from small disturbances. Furthermore, most of the inverse methods do not sufficiently consider the boundary conditions or structural topology, and these methods are limited to prior knowledge of precise structural loading or material, leading to a lack of potential for real-time SHM applications.

In Gherlone’s analysis of the shape sensing of structural deformation [14], Ko’s displacement method (Ko) [15], Modal Transformation Theory (MTT) [16], and the inverse finite element method (iFEM), [17] are currently the most popular methods for the real-time reconstruction of three-dimensional displacement by installing FBG sensors. In recent years, an increasing amount of literature has been published on iFEM [18]. This methodology can provide robust, stable, and accurate displacement results and is sufficiently fast for real-time monitoring applications, meeting the requirements of an SHM system for developing the next generation of aerospace vehicles. Comparatively, the iFEM is the innovative and seminal work of Tessler and Spangler et al. [17], who present the relationship between displacement/rotation degree of freedom and experimentally measured strain data in the inverse element, and the whole stiffness matrix can be sequentially assembled by the single inverse element matrix. Additionally, the geometrically complex structure is topologically divided into conventional plate, shell, and solid inverse elements.

The three-node inverse shell element (iMIN3) based on lowest-order anisoparametric C0 continuous and first-order shear deformation theory (FSDT) was thoroughly presented by Tessler et al. [17]. Afterward, the robust inverse-frame element was proposed by Gherlone et al. based on the assumptions of Timoshenko beam theory [18]. This method is tailored toward one-dimensional structures [19], such as trusses, beams, and frames subjected to stretching, bending, transverse-shear, and torsion-deformation. In the work of Roy et al. [20], the iFEM-beams analysis of the generic cross-section airfoil structure was significantly improved, and the reconstruction of wing test article structural deformations and loads was computationally conducted. Following the research of You and Ren et al. on inverse beam elements [21], iEBT2 was developed for the shape-sensing analysis of civil infrastructures, and the novel inclinometer using the iFEM theoretical framework was proposed for the deformation estimation of soil structures [22]. A remarkable effort has been made by Bao and co-workers to develop the enhanced inverse beam element. Zhao [23] coupled the isogeometric analysis approach to eliminate the multiple singularities problem due to discontinuities for beam structures, and Chen [24] presented a unified method for the reconstruction of Euler–Bernoulli and Timoshenko beams. To expand the practical usefulness of iFEM large-scale structures, Kefal proposed the new quadrilateral inverse-shell element (iQS4) [25], the eight-node curved inverse-shell element (iCS8) [26], and non-uniform rational B-spline (NURBS) technology [27]. Additionally, considerable research based on the iFEM family has been devoted to the possible applications of SHM for ships and marine platforms [26,28]. In addition, Niu presented the iMITC3+ based on the mixed interpolation of tensorial components technology for the reconstruction of thin plate structures [29], and Shang presents the iQAC using the quadrilateral area coordinates shape function to effectively deal with the element distortion [30]. Ji [31] proposed the deformation perception system by coupling the intelligent flexible sensing film (iFlexSense) and iFEM for the displacement reconstruction of a cantilever morphing aircraft compound.

Over the past decade, SHM systems have focused on the composite structure engineering field; however, the existing iFEM algorithms using FSDT cannot inadequately predict structures due to a high degree of anisotropy and heterogeneity. To solve this problem, Cerracchio [32] originally proposed the three-node iFEM/RZT for the reconstruction of composite sandwich plate structures based on the kinematic assumptions of the refined zigzag theory (RZT). Subsequently, Kefal [33] enhanced the reconstructed formulation by accurately redefining the first and second-order transverse shear strains, which conducted significantly accurate displacement predictions for thick multilayered composite and sandwich plate structures. Additionally, Sorrenti [34] developed the robust four-node quadrilateral iFEM/RZT free with shear locking for ultra-thin laminated composite plates, and the present method has been experimentally applied for the reconstruction of moderately thick wing-shaped sandwich structures by strain sensors [35].

Collectively, these studies outline the critical role iFEM plays in the displacement reconstruction of various types of plate structures; however, a small number of studies have focused on the transverse shear effect for the three-dimensional deformation field on thick plate structures. In addition, the main limitation of the existing iFEM, which is not available for thick plate structures, is that the shear displacement formulation for thick plate is not explicitly expressed, and the RZT requires the measurement strain information on the interface using embedded strain sensors. To address the above questions, research [36] has presented the enhanced iFEM, employing the scaled boundary finite element theory for reconstructing the thin plate structure using single strain information. The novel three-variable shear deformation plate formulation can effectively capture a shear deformable effect for thick plates [37].

To expand the library of existing iFEM for the SHM of practical engineering structures, this study aimed to develop a refined measurement method coupling the iFEM theoretical framework and analytical form for the shape analysis of thick plates. Furthermore, this work presents the reconstructed algorithm for the reconstruction of boundary surface displacement by employing strain information, and the methodology thoroughly shows the solution technology of the sixth-order differential equation to develop the deflection function.

This paper is organized as follows: Section 2 presents the theoretical framework of the refined iFEM formulation. Section 3 presents the different analysis models for thick plates and estimates the results from numerical and experimental analyses. Finally, the paper concludes with Section 4, which highlights the advantages of the present method.

## 2. Theory Formation

In this section, the order of the deformation equation of the plate structure under static loading is analytically derived in light of Hamilton’s principle. In addition, the iFEM element, using four-node element interpolation technology, is presented based on the variational principle and scaled boundary element methodology.

### 2.1. The Derivation Process of Displacement Function for Plate Mid-Plane

A three-dimensional linearly elastic, isotropic, homogeneous plate with the length a, width b, and thickness 2h in a Cartesian x,y,z coordinate system is considered (shown in Figure 1).

The three-dimensional displacement of arbitrary material points for plate structure can be written as.(1)$$\left\{\begin{array}{l} u_{x}(x,y,z)=-z\partial w_{b}/\partial y-z\left[5/3{\left(z/h\right)}^{2}-1/4\right]\partial w_{s}/\partial x\\ u_{y}(x,y,z)=-z\partial w_{b}/\partial y-z\left[5/3{\left(z/h\right)}^{2}-1/4\right]\partial w_{s}/\partial y\\ u_{z}(x,y,z)=w_{b}(x,y)+w_{s}(x,y) \end{array}\right.$$
where $\left(u_{x},u_{y},u_{z}\right)$ represent displacement components along x,y,z directions, $w_{b},w_{s}$ are the bending and shear components of transverse displacement on the mid-plane, respectively, and *h* is the plate thickness.

Following the research on the relation of shear and bending, the displacement can be defined as:(2)$$w_{s}=\alpha \nabla^{2}w_{b}$$
where $\nabla^{2}=\partial^{2}/\partial x^{2}+\partial^{2}/\partial y^{2}$ denotes the two-dimensional differential operator in the x,y directions, $\alpha =24h^{2}/\left(1-v)(15h-10\right)$ represents the proportion of shear displacement to the bending displacement, and can be calculated based on the geometry and material properties of the plate.

Substituting Equation (2) into Equation (1), the three-dimensional displacements can be expressed by two-dimensional mid-surface kinematic variables according to plate theory.(3)$$\left\{\begin{array}{l} u_{x}(x,y,z)=z\partial w_{b}/\partial x+\alpha g(z)\partial /\partial x\nabla^{2}w_{b}\\ u_{y}(x,y,z)=z\partial w_{b}/\partial y+\alpha g(z)\partial /\partial y\nabla^{2}w_{b}\\ u_{z}(x,y,z)=w_{b}+\alpha \nabla^{2}w_{b} \end{array}\right.$$
where $g(z)=5/3{\left(z/h\right)}^{3}-1/4z$ denotes the function along the z direction.

Based on the von Karman assumption, the strain–displacement relations can be expressed as follows:(4)$$\left\{\begin{array}{l} {\left[\begin{array}{ccc} \varepsilon_{xx} & \varepsilon_{yy} & \varepsilon_{xy} \end{array}\right]}^{T}={\bar{\varepsilon }}_{0}+z{\bar{\varepsilon }}_{1}+\alpha g(z){\bar{\varepsilon }}_{2}\\ {\left[\begin{array}{cc} \gamma_{xz} & \gamma_{yz} \end{array}\right]}^{T}=\alpha (1+\frac{dg}{dz}){\bar{\varepsilon }}_{s} \end{array}\right.$$
where$$\begin{array}{c} {\bar{\varepsilon }}_{1}=[\partial^{2}w_{b}/\partial x^{2},\;\;\;\partial^{2}w_{b}/\partial y^{2},\;\;\;2\partial^{2}w_{b}/\partial x\partial y]\;\;\;\;\;\;,\;\;\;\;\;\;{\bar{\varepsilon }}_{1}={\left[\partial^{2}/\partial x^{2},\;\;\;\partial^{2}/\partial y^{2},\;\;\;2\partial^{2}/\partial x\partial y\right]}^{T}\nabla^{2}w_{b}\;\;\;\;\;\;,\\ {\bar{\varepsilon }}_{s}=[\partial /\partial x,\;\;\;\;\partial /\partial Y]\nabla^{2}w_{b} \end{array}$$
${\bar{\varepsilon }}_{0}$ denote the in-plane membrane strains, ${\bar{\varepsilon }}_{1}$ the bending strain, ${\bar{\varepsilon }}_{2}$ the high-order strain, and ${\bar{\varepsilon }}_{s}$ the transverse-shear strains.

Based on the constitutive relationship between strain and stress, the internal forces can be expressed as:(5)$$\left\{\begin{array}{l} \left[{\bar{N}}_{xx},{\bar{N}}_{yy},{\bar{N}}_{xy}\right]={\bar{\varepsilon }}_{0}^{T}\bar{A}+{\bar{\varepsilon }}_{1}^{T}\bar{B}+{\bar{\varepsilon }}_{2}^{T}\bar{E}\\ \left[{\bar{M}}_{xx},{\bar{M}}_{yy},{\bar{M}}_{xy}\right]=\varepsilon_{0}^{T}\bar{B}+\varepsilon_{1}^{T}\bar{D}+\varepsilon_{2}^{T}\bar{F}\\ \left[{\bar{Q}}_{xz},{\bar{Q}}_{xz}\right]=\varepsilon_{s}^{T}D^{s} \end{array}\right.$$
where the coefficient can be expressed as:(6)$$\left\{\begin{array}{l} \left(\bar{A},\bar{B},\bar{D},\bar{E},\bar{F},\bar{H}\right)=\int_{-\frac{h}{2}}^{\frac{h}{2}}\left[1,z,z^{2},\alpha g,z\alpha g,\alpha^{2}g^{2}\right]C_{b}dz\\ {\bar{D}}^{s}=\int_{-\frac{h}{2}}^{\frac{h}{2}}\alpha^{2}{\left(1+\frac{dg}{dz}\right)}^{2}dz \end{array}\right.$$
$N_{,\lambda }\;\;(\lambda =xx,yy,xy),\;\;\;\;\;\;M_{,\lambda }\;\;(\lambda =xx,yy,xy)$ represent the forces and moments along the xy plane, respectively, and $Q_{,\gamma }\;\;(\gamma =xz,yz)$ represents the lateral shear force.

The higher-order forces, $P_{,\lambda }\;\;(\lambda =xx,yy,xy)$, take the form(7)$$[P_{xx},P_{yy},P_{xy}]=\int_{-\frac{h}{2}}^{\frac{h}{2}}\alpha g\left[\sigma_{xx},\sigma_{yy},\tau_{xy}\right]dz$$

According to the principle of virtual work, the energy equation of the entire structure can be expressed as:(8)δU+δV=0
where δU denotes the strain energy and δV denotes the work from external loads.

The work can be expressed as:(9)$$\delta V=-\int_{\Omega }\left\{q(x,y)(\delta w_{b}+\alpha \nabla^{2}\delta w_{b})\right.d\Omega$$

The strain energy can be expressed as:(10)$$\begin{array}{l} \delta U=\int_{V}\sigma \delta \varepsilon dV\\ \;\;\;\;\;\;\;\;=\int_{\Omega }\left\{[N_{xx},\;\;\;\;N_{yy},\;\;\;\;N_{xy}]\delta \varepsilon_{0}+[M_{xx},\;\;\;\;M_{yy},\;\;\;\;M_{xy}]\delta {\bar{\varepsilon }}_{1}\right.\\ \;\;\;\;\;\;\;\;\;\;\;\;\;\;+\;[P_{xx}\;\;\;\;P_{yy}\;\;\;\;P_{xy}]\delta \varepsilon_{2}+\left.[Q_{xz}\;\;\;\;Q_{yz}]\delta \varepsilon_{s}\right\}d\Omega \end{array}$$

According to the force balance relationship of the plate structure, the three displacement variables can be written as:(11)$$\left\{\begin{array}{l} \frac{\partial }{\partial x}({\bar{A}}_{11}\frac{\partial u_{0}}{\partial x}+{\bar{A}}_{12}\frac{\partial v_{0}}{\partial y}-{\bar{B}}_{11}\frac{\partial^{2}w_{b}}{\partial x^{2}}-{\bar{B}}_{12}\frac{\partial^{2}w_{b}}{\partial y^{2}}+{\bar{E}}_{11}\frac{\partial^{2}\nabla^{2}w_{b}}{\partial x^{2}}+{\bar{E}}_{12}\frac{\partial^{2}\nabla^{2}w_{b}}{\partial y^{2}})+\frac{\partial }{\partial y}[{\bar{A}}_{66}(\frac{\partial u_{0}}{\partial y}+\frac{\partial v_{0}}{\partial x})-2{\bar{B}}_{11}\frac{\partial^{2}w_{b}}{\partial x\partial y}+2{\bar{E}}_{66}\frac{\partial^{2}\nabla^{2}w_{b}}{\partial x\partial y})=0\\ \frac{\partial }{\partial x}({\bar{A}}_{66}(\frac{\partial u_{0}}{\partial y}+\frac{\partial v_{0}}{\partial x})-2{\bar{B}}_{66}\frac{\partial^{2}w_{0}}{\partial x\partial y}+2{\bar{E}}_{66}\frac{\partial^{2}\nabla^{2}w_{b}}{\partial x^{2}}+{\bar{E}}_{12}\frac{\partial^{2}\nabla^{2}w_{b}}{\partial y^{2}})+\frac{\partial }{\partial y}[{\bar{A}}_{12}\frac{\partial u_{0}}{\partial x}+{\bar{A}}_{22}\frac{\partial v_{0}}{\partial y}-{\bar{B}}_{12}\frac{\partial^{2}w_{b}}{\partial x^{2}}-{\bar{B}}_{22}\frac{\partial^{2}w_{b}}{\partial y^{2}}+{\bar{E}}_{12}\frac{\partial^{2}\nabla^{2}w_{b}}{\partial x^{2}}+{\bar{E}}_{22}\frac{\partial^{2}\nabla^{2}w_{b}}{\partial y^{2}})=0\\ \frac{\partial^{2}}{\partial x^{2}}({\bar{B}}_{11}\frac{\partial u_{0}}{\partial x}+{\bar{B}}_{12}\frac{\partial u_{0}}{\partial x}-{\bar{D}}_{11}\frac{\partial^{2}w_{0}}{\partial x^{2}}-{\bar{D}}_{12}\frac{\partial^{2}w_{b}}{\partial y^{2}}+{\bar{F}}_{11}\frac{\partial^{2}w_{b}}{\partial x^{2}}-{\bar{F}}_{12}\frac{\partial^{2}w_{b}}{\partial y^{2}})+2\frac{\partial^{2}}{\partial x\partial y}({\bar{B}}_{66}(\frac{\partial u_{0}}{\partial y}+\frac{\partial v_{0}}{\partial x})-2{\bar{D}}_{66}\frac{\partial^{2}w_{b}}{\partial x\partial y}+{\bar{F}}_{66}\frac{\partial^{2}w_{b}}{\partial x\partial y})\\ +\frac{\partial^{2}}{\partial y^{2}}({\bar{B}}_{12}\frac{\partial u_{0}}{\partial x}+{\bar{B}}_{22}\frac{\partial v_{0}}{\partial y}-{\bar{D}}_{12}\frac{\partial^{2}w_{b}}{\partial x^{2}}-{\bar{D}}_{22}\frac{\partial^{2}w_{b}}{\partial y^{2}}+{\bar{F}}_{12}\frac{\partial^{2}w_{b}}{\partial x^{2}}-{\bar{F}}_{22}\frac{\partial^{2}w_{b}}{\partial y^{2}})+q(x,y)=0 \end{array}\right.$$

For the isotropic elastic plate structure, Equation (11) is simplified as:(12)$$-\bar{D}\nabla^{4}w_{b}+\bar{F}\nabla^{6}w_{b}+q(x,y)=0$$
where, $\nabla^{4}=\partial^{4}/\partial x^{4}+2\partial^{4}/\partial x^{2}\partial y^{2}+\partial^{4}/\partial y^{4},\;\;\;\;\;\;\;\;\;\;\nabla^{6}=\partial^{6}/\partial x^{6}+3\partial^{6}/\partial x^{4}\partial y^{2}+3\partial^{6}/\partial x^{2}\partial y^{4}+\partial^{6}/\partial y^{6}$, $\bar{D}=Eh^{3}/12(1-v^{2})$ is the constant related to the material properties, $\bar{F}=\alpha E/1-v^{2}\int_{-\frac{h}{2}}^{\frac{h}{2}}zg(z)dz$ is the constant related to the integration of cross-sectional deformation function along the thickness direction, and q(x,y) is the external load.

### 2.2. The Order of Mid-Plane Deformation Function in Thick Plate Elements

Without considering the effect of external loads, the equation is simplified as follows:(13)$$\frac{\partial^{6}w_{b}}{\partial x^{6}}+3\frac{\partial^{6}w_{b}}{\partial x^{4}\partial y^{2}}+3\frac{\partial^{6}w_{b}}{\partial x^{2}\partial y^{4}}+\frac{\partial^{6}w_{b}}{\partial y^{6}}+c(\frac{\partial^{4}w_{b}}{\partial x^{4}}+2\frac{\partial^{4}w_{b}}{\partial x^{2}\partial y^{2}}+\frac{\partial^{4}w_{b}}{\partial y^{4}})=0$$
where the constant $c=\bar{F}/\bar{D}$ can be defined by material property.

According to the method for solving fourth-order deflection partial differential equations, the deflection function can be expressed using the separation of variables method as follows:(14)$$w_{b}(x,y)=\psi (x)\phi (y)$$
where ψ(x),ϕ(y) represents the deflection function of the neutral plane along the x and y directions, respectively, and can be expressed as:(15)$$\psi (x)=e^{ux},\;\;\;\;\;\;\;\;\;\;\;\;\;\;\;\;\;\;\;\;\;\;\;\phi (y)=e^{\lambda y}$$
where u,λ denotes the spectral parameters of the deflection function along the x and y directions, respectively.(16)$$u^{6}+3u^{2}\lambda^{4}+3u^{4}\lambda^{2}+\lambda^{6}+c(u^{4}+2u^{2}\lambda^{2}+\lambda^{4})=0$$

According to the assumptions in Equation (14), the spectral parameters u,λ along x and y directions are independent of each other, and the method of fixing a single variable can be used for solving the parameters.

Assuming that the *y*-direction spectral parameters are known quantities, the solution in Equation (16) can be expressed as:(17)$$u_{1,2}=\pm \sqrt{\lambda^{2}+c},\;\;\;\;\;\;\;\;\;u_{3,4}=\pm \sqrt{\lambda^{2}+c},\;\;\;\;u_{5,6}=\pm \sqrt{\lambda^{2}}$$
where $u_{1,2},\;\;\;u_{3,4},\;\;\;u_{5,6}$ denote the eigenvector solution. Similarly, the eigenvector solution along the x-direction can be obtained as follows:(18)$$\lambda_{1,2}=\pm \sqrt{u^{2}+c},\;\;\;\;\;\;\;\;\;\lambda_{3,4}=\pm \sqrt{u^{2}+c},\;\;\;\;\lambda_{5,6}=\pm \sqrt{u^{2}}$$

By replacing the solution along the *x*, *y*-direction, the deflection function ψ(x),ϕ(y) can be expressed, respectively, as:(19)$$\left\{\begin{array}{l} \psi (x)=A_{1}\cosh\alpha_{1}x+A_{2}\sinh\alpha_{1}x+B_{1}\cosh\alpha_{2}x+B_{2}\sinh\alpha_{2}x\\ \;\phi (y)=C_{1}\cosh\beta_{1}y+C_{2}\sinh\beta_{1}y+D_{1}\cosh\beta_{2}y+D_{2}\sinh\beta_{2}y \end{array}\right.$$
where $A_{1},A_{2},B_{1},B_{2},C_{1},C_{2},D_{1},D_{2}$ denote the unknown quantity, respectively, cosh,sinh denote the double sine and cosine symbols, and $\alpha_{1}=\;\sqrt{\lambda^{2}+c}$, $\;\alpha_{2}=\;\sqrt{\lambda^{2}}$, $\beta_{1}=\;\sqrt{u^{2}+c}$, $\beta_{2}=\;\sqrt{u^{2}}$ denote the spectral parameters.

According to the principle of double sine-cosine Taylor series expansion, taking the first three terms of the cosine Taylor series, the deflection function along the *x*, *y*- direction can be expressed as:(20)$$\left\{\begin{array}{l} \psi (x)=c_{0}+c_{1}x+c_{2}x^{2}+c_{3}x^{3}+c_{4}x^{4}+c_{5}x^{5}\\ \phi (y)=d_{0}+d_{1}y+d_{2}y^{2}+d_{3}y^{3}+d_{4}y^{4}+d_{5}y^{5}\; \end{array}\right.$$

By taking the product of Equation (20), the deflection function can be presented as:(21)w(x,y)=CC•RR(x,y)
where $RR=[1,x,y,\ldots x^{4}y^{5},x^{5}y^{4},x^{5}y^{5}]$ indicates the base function of quadrilateral elements and can be selected by the Pascal polynomial. $CC=[a_{1},a_{2},a_{3},\ldots \ldots a_{36}]$ is the coefficient vector on the base function RR.

### 2.3. Mathematical Framework of iFEM

This section introduces the iFEM based on the scaled boundary method [37], which obtained the structural displacements of the top/bottom surfaces based on single surface strain information. According to the study on iFEM based on the scaled boundary element theory, the element reconstructed equation can be presented as:(22)$$k^{e}u^{e}=f^{e}$$
where $\;\;k^{e}$ is the element stiffness matrix with respect to the deployment of strain sensors, $\;\;f^{e}$ is the similar load vec tor on the discrete strain sensor’s reading, and $\;\;u^{e}=\left\{u_{x}^{i},\;\;\;\;\;u_{y}^{i},\;\;\;\;u_{z}^{i}\right\}$ is the nodal displacement vector of the element.

The element matrix $\;k^{e}$ can be explicitly expressed b:(23)$$k^{e}=\sum_{i=1}^{n}B^{m}{\left(x_{i},y_{i}\right)}^{T}m_{i}(\theta_{i})B^{m}(x_{i},y_{i})+\sum_{i=1}^{n}B^{s}{\left(x_{i},y_{i}\right)}^{T}B^{s}(x_{i},y_{i})$$

$\;f^{e}$ can be written as:(24)$$f^{e}=\sum_{i=1}^{n}B^{m}{\left(x_{i},y_{i}\right)}^{T}\cdot m_{i}(\theta_{i})\cdot \overset{⌢}{\varepsilon }(x_{i},y_{i})+\sum_{i=1}^{n}B^{s}{\left(x_{i},y_{i}\right)}^{T}\cdot {\overset{⌢}{g}}_{i}(x_{i},y_{i})$$

### 2.4. Computation of Experimental Measurement Strains

In the iFEM formulation, the deployment of strain sensors, such as strain rosettes and FBG sensors, is crucial for dynamically reconstructing structural deformation. Different from the strain deployment based on the existing iFEM (shown in Figure 2a), the enhanced iFEM only requires the strain information on one surface of the structure (top or bottom surfaces are available), and the mounted deployment of strain sensors is shown in Figure 2b.

In Tessler’s analysis of weighting constants on iFEM [17], the appropriate applications of weighting constants can enable an inverse element using very sparse measured strain data. For the coefficient $\;w_{e}$, as the strains of analytical and corresponding experimental readings are obtained, $\;w_{e}$ will be equal to unity. For the elements without experimentally measured data, the coefficient is assigned the value of $10^{-3}\sim 10^{-6}$. In addition, the coefficient $w_{g}$ on the second least-squares term will be assigned $10^{-5}\sim 10^{-7}$ for reconstructing the deformation of thin or moderately thick structures.

Following Equation (22), the three-dimensional displacement field on the boundary surface is recovered in real-time. Subsequently, a quantitative relationship between surface deformation and the coefficients of mid-surface function  w(x,y) is built, expressed as(25)$$U^{e}=Hc$$
where $\;U^{e}={\left[u_{ix}^{e},u_{iy}^{e},u_{iz}^{e}\right]}^{T}$ is the nodal displacement, H is the function matrix related to nodal global coordinates, and $c=[a_{1},a_{2},a_{3},\ldots \ldots a_{36}]$ is the unknown coefficient of mid-plane function.

Introducing Equation (22) into Equation (25), the coefficient *c* can be calculated and the three-dimensional deformation along the thickness can be recovered using a defined mid-surface function.

## 3. Numerical and Experimental Validation

The thick plate structures are analyzed in this section to demonstrate iFEM’s capability for reconstructing the deformation of thick plate models. In addition, compared with the reference value from the direct finite element method (FEM) analysis, the accuracy of iFEM is quantitatively assessed using the indices of root mean square error (RMS), percentage difference (PD), and the maximum root mean square error percentage (MRMS).

RMS, PD, and MRMS can be estimated as follows:(26)$$\left\{\begin{array}{l} RMS=\sqrt[{2}]{\sum_{i=1}^{n}{\left(w_{i}^{iFEM}(x_{i},y_{i})-w_{i}^{ref}(x_{i},y_{i})\right)}^{2}/n}\\ PD(w(i))=100\%\times \left|\frac{w^{ref}(i)-w^{\exp}(i)}{w^{\exp}(i)}\right|\\ MRMS=100\%\times \frac{RMS}{w_{\max}} \end{array}\right.$$
where $\;\;w_{i}^{ref}$ indicates the benchmark results through the transverse direction and can be calculated using analytical software or experimentally measured equipment, $w_{i}^{iFEM}$ represents the results reconstructed from the iFEM, *n* indicates the number of sampling points, and $w_{\max}$ represents the maximum deformation amplitude for different loading conditions.

### 3.1. Numerical Examples

#### 3.1.1. The Model-I for Cantilever Rectangular Plate

As depicted in Figure 3, the rectangular plate under cantilevered constraint is reconsidered, with a length of a=1.5 m, height of b=0.5 m, and uniform thickness of 100 mm. The thickness/width ratio is 5, which is classified as a thick plate structure.

The plate comprises aluminum with Young’s modulus E=73 GPa and Poisson’s ratio v=0.3. The material properties are used to provide the input strain reading and compare the deformation calculated by the finite element model.

The plate is subjected to displacement constraint along the z-direction at a location 15 mm from the free tip. To assess the accuracy of the iFEM, the direct finite element plate model is meshed through high-fidelity elements. The discretization of the whole structure using inverse elements based on iFEM is shown in Figure 4a, and the deployment of the FBG strain sensor is shown in Figure 4a.

The structure is divided into 10 inverse elements, and four measured points (single-direction strain data) are suitably mounted on each element. The locations of these measured points can be obtained using the optimal model.

As can be seen in Table 1, the maximum amplitude under the bending response along the z-direction is 50 mm, the maximum error value is 1.3 mm, and the maximum present difference is 2.5%. The displacement contour plots on the mid-plane along the z-direction are shown in Figure 5. Figure 5a depicts the result of the direct FEM, Figure 5b shows the reconstructed deformation error plot based on the present iFEM; the maximum error along the z-direction is 1.3 mm.

Figure 6 shows the deformation contour plots of the mid-plane of the plate structure under the simultaneous application of displacement constraints of −50 mm, −50 mm, and 10 mm along the z-direction at points 1, 2, and 3.

The maximum deformation along the positive direction is 49.10 mm, and the maximum deformation along the negative direction is 51.10 mm. According to the reconstructed deformation error cloud map presented in Figure 6b, the maximum errors along the positive and negative directions are 1.35 mm and −1.53 mm, respectively, with a maximum reconstruction error percentage of 2.9%.

Table 2 shows the deformation reconstruction results of the surface sampling points along the *z*-direction on the rectangular thick plate structure. The maximum deformation along the *z*-direction is 41.80 mm, the maximum error is 1.50 mm, and the reconstructed *RMS* value is 0.85 mm.

Figure 7 shows the deformation contour plots of the plate structure mid-plane under multi-point loading. The maximum deformation along the *y*-direction is 10.1 mm, with a maximum error of 0.45 mm and a maximum reconstruction error percentage of 4.5%. Table 3 shows the reconstructed results of the sampling points along the y-direction. The maximum deformation along the *y*-direction is 9.66 mm, the maximum error is 0.41 mm, and the reconstructed *RMS* value is 0.26 mm.

#### 3.1.2. The Model-II for Cantilever Wing Shape Plate

To demonstrate the universality on iFEM for thick plate structures, a wing-shaped plate structure under a cantilevered configuration is analyzed as an example (shown in Figure 8a).

The wing-shaped plate has a length of L=1200 mm. The root chord b=300 mm, the tip chord c=200 mm, the thickness b=20 mm, and the angle of the leading edge θ=15°, respectively. The material properties of the structure are quantitated using Young’s modulus $E=0.73\times 10^{5}\;\mathrm{Mpa}$, Poisson’s ratio v=0.3, and mass density $\rho =2712.63\;\mathrm{kg}/m^{3}$. The finite element model can be characterized by high-fidelity discretization quadrilateral meshes. The structure is discretized into four inverse elements (shown in Figure 8b), and four different FBG strain sensors are installed along the *x*-direction in one inverse element. A displacement constraint of 75 mm is applied along the *z*-direction at point 1.

According to the reconstruction algorithm, the displacement deformation of samples on the surface along the x-direction is listed in Table 4. Considering the deformation analyzed using the FEM as the benchmark, the maximum deformation is 4.00 mm, the maximum error is 0.19 mm, and the *RMS* value is 0.14 mm.

The displacement deformation contour plot of the mid-plane along the x-direction is shown in Figure 9a, with a maximum deformation of 3.9 mm. Figure 8b shows the displacement deformation error contour plot reconstructed along the *x*-direction, with a maximum error of 0.2 mm, and a maximum reconstruction error percentage of 5.1%.

Table 5 shows the reconstruction results of displacement deformation along the z-direction of the samples on the surface, with a maximum deformation of 65.10 mm, a maximum error of 2.08 mm, and a reconstructed RMS value of 1.19 mm.

The deformation contour plot along the z-direction of the mid-plane of the thick plate structure is shown in Figure 10a, with a maximum variable of 73.0 mm. The reconstruction deformation error plot along the direction of the mid-plane of the thick plate structure is shown in Figure 10b, with a maximum error of 2.822 mm and a maximum reconstruction percentage error of 3.8%.

The present analysis demonstrated the potential capability of iFEM for the shape sensing of a thick plate structure based on single strain information. Furthermore, these numerical cases confirm the high accuracy of the iFEM in the real-time reconstruction of general and special function plate structures.

### 3.2. Experimental Validation

To verify the effectiveness of the application of plate structure, the wing-shaped plate (shown in Figure 11) has a length of L=700 mm. The root chord b=200 mm, the tip chord c=80 mm, and the thickness h=5 mm, respectively. The material properties of the structure are quantitated using Young’s modul us E=7.3 Gpa, Poisson’s ratio v=0.3, and mass density ρ=2712.63 kg/m.

According to the geometric dimensions of the plate structure, the whole structure is discretized with three inverse elements and four FBG strain sensors (including two 0-angle, one 45-angle, one 90-angle FBG strains along the *x*-direction) are optimally installed in the one inverse element. To examine the accuracy of this method for different degrees of deformation amplitude, the experimental loading setup is presented in Figure 12a, and three types of force are produced on the loading point 30 mm away from the right boundary.

The real deformation state of the cantilever plate can be represented by the number of discrete markers pasted on the surface (shown in Figure 12b), and the three-dimensional coordinates of the markers can be acquired using NDI (shown in Figure 12c). The real-time strain information can be collected by the FBG demodulator device (shown in Figure 12d).

By applying two different forces on force points along the z-direction, the reconstructed deformation and experimental measurement results (selecting the 10 markers) from NDI can be obtained, respectively. Considering the deformation analyzed using the experimental measurement system as a benchmark, the displacement deformation of nine marker samples on the surface along the z-direction is listed in Table 6 and Table 7, respectively.

For loading one (shown in Table 6), the maximum deformation is 20.84 mm, the maximum error is 2.21 mm, and the *RMS* value is 1.37 mm, with an MRMS of 6.5%. For loading two (shown in Table 7), the maximum deformation is 43.75 mm, the maximum error is 4.00 mm, and the *RMS* value is 2.41 mm, with an MRMS of 6.5%. Comparing the accuracy index of MRMS based on model-II with the value of 1.67% along the z-direction for the numerical analysis, the effect is lower than the numerical analysis value, which can be ascribed to unavoidable measurement errors, such as the structural assembly and strain measurement errors.

According to these results from the experimental measurement, the assessment index value of MRMS completely satisfies the project requirements, and the present analysis demonstrates the potential capability of inverse elements to shape-sense geometrically complex plate structures. Furthermore, the experimental analysis results confirmed that the method provides innovative insights into SHM field systems of irregular shape structures based on sparse strain sensor deployment.

## 4. Conclusions

This study presents an improved shape sensing method that couples the analytical formulation with the inverse finite element theoretical framework and is extensively applicable to reconstructing the full-field displacement of thick plate structures based on single-surface strain information. Additionally, this finding expands the engineering applications of SHM systems for laminate composite structures.

The reconstructed algorithm presents the bending deformation differential equation based on the relation between bending and transverse shear deformation on the mid-plane. The analytical form of the bending function is derived using the separation-of-variable method. In addition, the multiple parameters of the bending function are obtained by employing the inverse finite element method based on the scaled boundary element, and the deformation field on the mid-plane can be reconstructed using the discrete measurement strain.

Furthermore, the different numerical model results comparing FEM analyses demonstrate that the enhanced algorithm has better performance in reconstructing the displacement field of thick plate structures, and the effectiveness of the method is evaluated using the experimental wing plate. The present study fills a gap in the existing research on iFEM for engineering applications of plate/shell structures.

## Figures and Tables

**Figure 1 micromachines-16-00368-f001:**
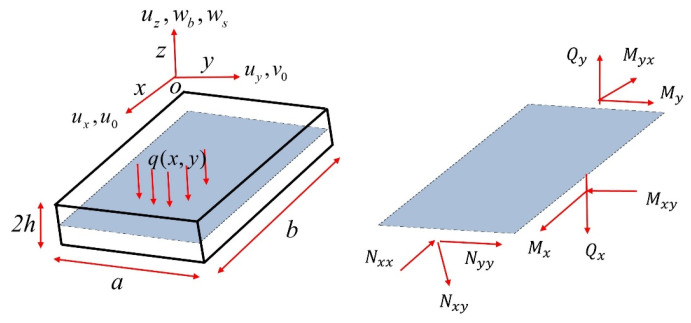
The plate element.

**Figure 2 micromachines-16-00368-f002:**
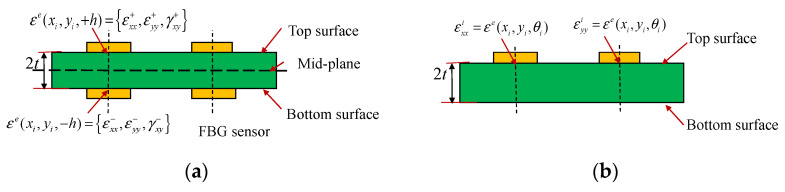
Discrete deployment of surface strains measured by FBG sensors in inverse element at discrete locations; (**a**) The existing iFEM, (**b**) The refined iFEM.

**Figure 3 micromachines-16-00368-f003:**
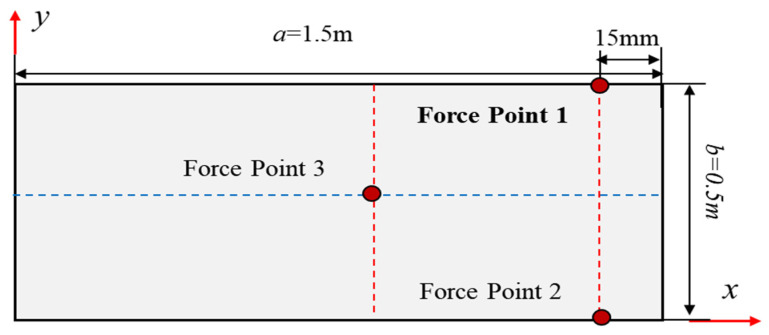
Cantilever plate under transverse force applied near the free tip.

**Figure 4 micromachines-16-00368-f004:**
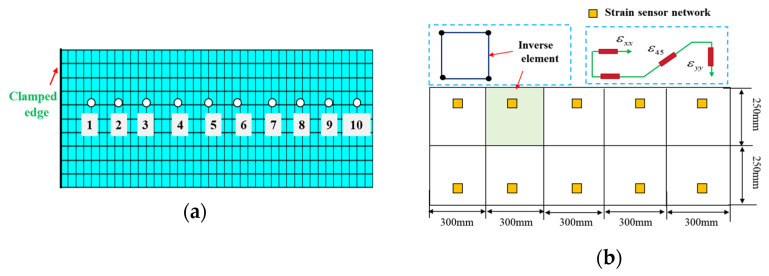
The cantilever plate element discretization: (**a**) FEM, (**b**) iFEM using 10 inverse elements and four single strains per element.

**Figure 5 micromachines-16-00368-f005:**
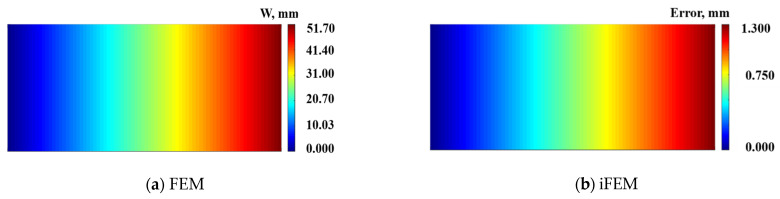
Contour plots of W displacement on neutral plane along the z-direction: (**a**) Direct FEM analysis; (**b**) Error plot based on iFEM.

**Figure 6 micromachines-16-00368-f006:**
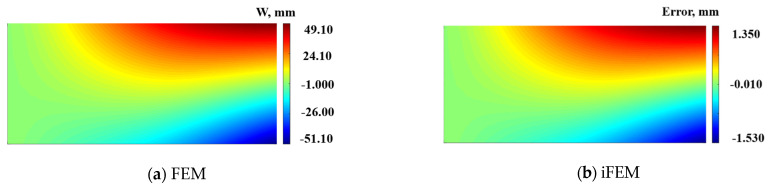
Contour plots of W displacement on neutral plane along the z-direction: (**a**) Direct FEM analysis. (**b**) Error plot based on iFEM.

**Figure 7 micromachines-16-00368-f007:**
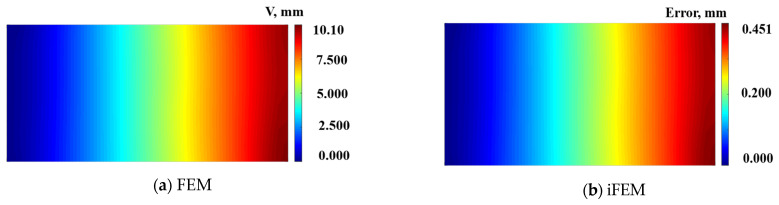
Contour plots of W displacement on mid-plane along the y-direction: (**a**) Direct FEM analysis. (**b**) Error plot based on iFEM.

**Figure 8 micromachines-16-00368-f008:**
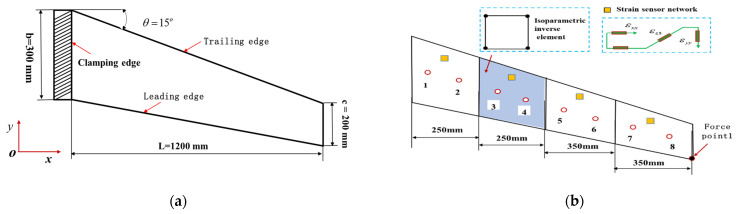
The wing-shaped plate model under cantilevered configuration: (**a**) The geometric dimension of plate structure; (**b**) The inverse element and high-fidelity finite elements discretization.

**Figure 9 micromachines-16-00368-f009:**
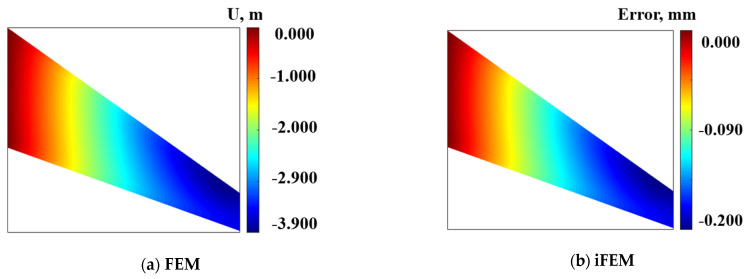
Contour plots of U displacement along x-direction; (**a**) Direct FEM analysis, (**b**) Error plot based on iFEM.

**Figure 10 micromachines-16-00368-f010:**
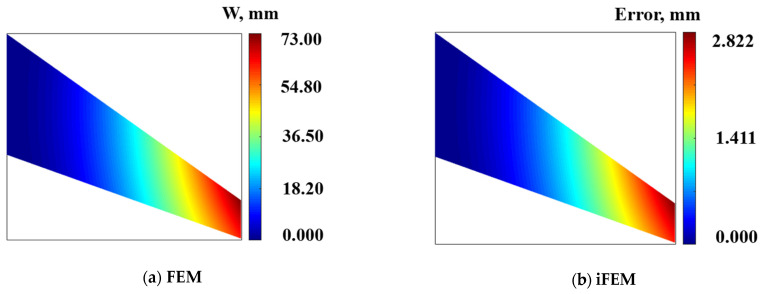
Contour plots of displacement along z-direction: (**a**) Direct FEM analysis. (**b**) Error plot based on iFEM.

**Figure 11 micromachines-16-00368-f011:**
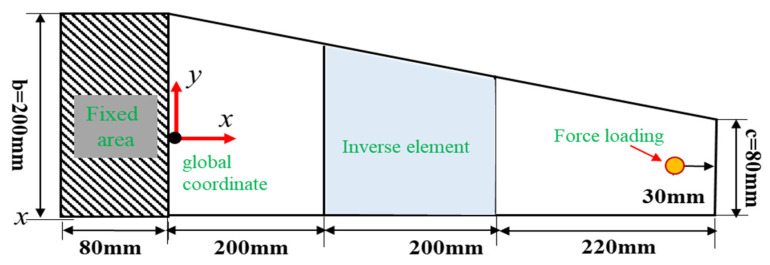
The geometric dimension of experimental wing plate structure and the inverse element.

**Figure 12 micromachines-16-00368-f012:**
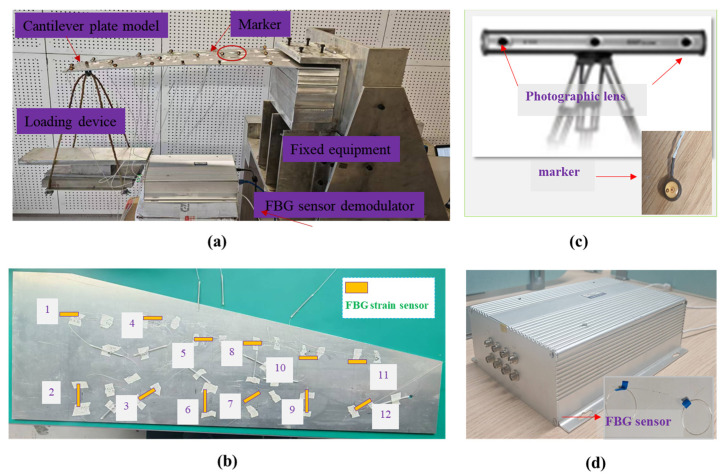
Experimental setup of wing plate: (**a**) Wing plate deformation under loading; (**b**) The layout of FBG strain sensors on one surface; (**c**) NDI measurement system for markers’ three-dimensional coordinates; (**d**) FBG sensor demodulator for FBG strain sensor.

**Table 1 micromachines-16-00368-t001:** The deformation of sampling points on the plate surface along the z-direction.

Sampling Points	Benchmark Value (mm)	Reconstruct Value (mm)	Error Value (mm)
1	8.62	8.17	0.45
2	13.79	13.27	0.52
3	17.24	16.68	0.56
4	22.41	21.68	0.73
5	27.59	26.80	0.79
6	32.76	31.70	1.06
7	36.21	35.04	1.17
8	39.66	38.50	1.16
9	44.83	43.63	1.20
10	50.00	48.70	1.30

**Table 2 micromachines-16-00368-t002:** The deformation of sampling points on the plate surface along the z-direction.

Sampling Points	Benchmark Value (mm)	Reconstruct Value (mm)	Error Value (mm)
1	1.94	1.75	0.19
2	4.18	3.56	0.62
3	5.78	5.08	0.70
4	7.94	7.02	0.92
5	9.36	8.42	0.94
6	9.37	8.44	0.93
7	8.39	7.58	0.81
8	6.90	6.01	0.89
9	41.80	40.30	1.50
10	0.64	0.40	0.24

**Table 3 micromachines-16-00368-t003:** The deformation of sampling points on the plate surface along the y-direction.

Sampling Points	Benchmark Value (mm)	Reconstruct Value (mm)	Error Value (mm)
1	1.15	1.00	0.15
2	2.18	1.93	0.25
3	2.91	2.70	0.21
4	4.00	3.73	0.27
5	5.11	4.92	0.19
6	6.21	5.92	0.29
7	6.94	6.64	0.30
8	7.67	7.47	0.20
9	6.61	6.30	0.31
10	9.66	9.25	0.41

**Table 4 micromachines-16-00368-t004:** The deformation of sampling points on the plate surface along the x-direction.

Sampling Points	Benchmark Value (mm)	Reconstruct Value (mm)	Error Value (mm)
1	−1.12	−1.07	0.05
2	−1.71	−1.63	0.08
3	−2.40	−2.26	0.14
4	−3.00	−2.85	0.15
5	−3.39	−3.24	0.15
6	−3.71	−3.53	0.18
7	−3.74	−3.55	0.19
8	−4.00	−3.79	0.21

**Table 5 micromachines-16-00368-t005:** The deformation of sampling points on the plate surface along the z-direction.

Sampling Points	Benchmark Value (mm)	Reconstruct Value (mm)	Error Value (mm)
1	5.00	4.82	0.18
2	10.80	10.43	0.37
3	18.60	18.02	0.58
4	25.50	24.64	0.86
5	35.80	34.68	1.12
6	49.80	48.00	1.80
7	65.10	63.02	2.08
8	70.50	68.35	2.15

**Table 6 micromachines-16-00368-t006:** The deformation of sampling points along the z-direction for loading one.

Marker	Benchmark Value (mm)	Reconstruct Value (mm)	Error Value (mm)
1	1.162	0.912	0.250
2	2.573	2.123	0.450
3	4.237	3.437	0.800
4	7.330	6.43	0.900
5	9.659	8.639	1.020
6	12.037	10.537	1.500
7	15.415	13.615	1.800
8	18.556	16.626	1.930
9	20.838	18.628	2.210

**Table 7 micromachines-16-00368-t007:** The deformation of sampling points along the z-direction for loading two.

Marker	Benchmark Value (mm)	Reconstruct Value (mm)	Error Value (mm)
1	2.789	2.289	0.500
2	7.204	6.584	0.620
3	12.712	11.412	1.300
4	16.273	14.773	1.500
5	23.181	21.181	2.000
6	27.204	25.104	2.100
7	33.297	30.397	2.900
8	43.049	39.199	3.850
9	43.759	39.759	4.000

## Data Availability

The original contributions presented in this study are included in the article. Further inquiries can be directed to the corresponding author(s).

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
