# Peer review of "The Enhanced Measurement Method Based on Fiber Bragg Grating Sensor for Structural Health Monitoring"

_micromachines, 2025, doi:10.3390/mi16040368_

Round 1
Reviewer 1 Report
Comments and Suggestions for Authors
The manuscript presents a valuable advancement in iFEM-based shape sensing by incorporating Fiber Bragg Grating strain sensors to address transverse shear limitations in thick plate structures. The proposed method effectively expands the applicability of SHM systems, and the numerical validation demonstrates its accuracy under different loading conditions. This work provides important insights and has strong potential for practical implementation in structural monitoring. I have one question: how can the proposed method address geometric anomalies, such as cracks located inside the thick plate that are invisible from the surface?
Author Response
Dear Reviewer :
We have submitted the revised manuscript " The Enhanced Measurement Method based on Fiber Bragg Grating Sensor for Structural Health Monitoring" (micromachines-3506621). We have carefully studied the reviewers’ comments and made the corresponding revisions. These comments are really valuable and helpful for improving our manuscript. In the revised manuscript, the revised parts have been marked in red.
Our responses to the reviewers’ comments are described in the following pages, where these comments are presented in black and the authors' responses are given in red.
We appreciate you for all your work. We are looking forward to hearing from you.
With best regards
Shengtao Niu, Ru Li
Appendix:
Thank you for your careful review of our manuscript, the responses to your questions are described in the following PDF file.

Reviewer 2 Report
Comments and Suggestions for Authors
The article integrates the analytical formula with the iFEM theoretical framework to clarify the relationship between bending and transverse shear displacement fields, proposing a method that reconstructs three-dimensional deformation of thick plates using only single-surface FBG strain data. This approach demonstrates uniqueness in existing FBG-based sensing technologies. The established sixth-order differential equations and variational solving method exhibit theoretical depth, expanding the applicability of iFEM in composite materials. The following limitations and unresolved questions require further investigation in this study.
1. The current verification of the method is limited to numerical simulations under static load conditions, lacking experimental data support. Laboratory or field tests should be supplemented to compare reconstructed results with measured data, confirming reliability in real-world scenarios.
2. While the article mentions environmental interference in non-contact measurements (e.g., measurement scenes and surrounding environments), similar issues exist for contact sensors. Further analysis of temperature compensation for FBG strain measurements is recommended to enhance engineering applicability.
3. Annotation errors:
Line 298: The coefficient "cc" should be corrected to "c".
Figure 6: The first occurrence of Figure 6 duplicates Figure 5 and should be deleted.
Figures 5–7: Missing deformation direction labels (e.g., along x/y/z-direction). A thorough review of all directional annotations is required to avoid inconsistencies.
4. The computational efficiency of the algorithm is not explicitly quantified. Clarify its real-time monitoring capabilities, including dynamic monitoring potential.
Author Response

(The authors gave the same response as above.)
